# Time-Restricted Feeding Ameliorates Methionine–Choline Deficient Diet-Induced Steatohepatitis in Mice

**DOI:** 10.3390/ijms25031390

**Published:** 2024-01-23

**Authors:** Ik-Rak Jung, Rexford S. Ahima, Sangwon F. Kim

**Affiliations:** Department of Medicine, Division of Endocrinology, Diabetes and Metabolism, Johns Hopkins University, Baltimore, MD 21218, USA; ijung4@jhmi.edu

**Keywords:** inositol polyphosphate multikinase (IPMK), time-restricted feeding (TRF), methionine–choline deficient diet (MCDD), non-alcoholic steatohepatitis (NASH)

## Abstract

Non-alcoholic steatohepatitis (NASH) is an inflammatory form of non-alcoholic fatty liver disease (NAFLD), closely associated with disease progression, cirrhosis, liver failure, and hepatocellular carcinoma. Time-restricted feeding (TRF) has been shown to decrease body weight and adiposity and improve metabolic outcomes; however, the effect of TRF on NASH has not yet been fully understood. We had previously reported that inositol polyphosphate multikinase (IPMK) mediates hepatic insulin signaling. Importantly, we have found that TRF increases hepatic IPMK levels. Therefore, we investigated whether there is a causal link between TRF and IPMK in a mouse model of NASH, i.e., methionine- and choline-deficient diet (MCDD)-induced steatohepatitis. Here, we show that TRF alleviated markers of NASH, i.e., reduced hepatic steatosis, liver triglycerides (TG), serum alanine transaminase (ALT) and aspartate aminotransferase (AST), inflammation, and fibrosis in MCDD mice. Interestingly, MCDD led to a significant reduction in IPMK levels, and the deletion of hepatic IPMK exacerbates the NASH phenotype induced by MCDD, accompanied by increased gene expression of pro-inflammatory chemokines. Conversely, TRF restored IPMK levels and significantly reduced gene expression of proinflammatory cytokines and chemokines. Our results demonstrate that TRF attenuates MCDD-induced NASH via IPMK-mediated changes in hepatic steatosis and inflammation.

## 1. Introduction

Non-alcoholic steatohepatitis (NASH) is a severe progression of non-alcoholic fatty liver disease (NAFLD), characterized by hepatic triglyceride (TG) accumulation, inflammation, liver fibrosis, and hepatocellular injury, which can progress to cirrhosis and hepatocellular carcinoma (HCC) [1]. Numerous studies have proposed mediators that regulate NASH initiation and progression, including insulin resistance [2], oxidative stress [3], mitochondrial dysfunction [4], cell death [5], immune cell activation [6], and genetic/epigenetic factors [7]. Notably, there is evidence indicating a potential link between insulin resistance and an elevated risk of hepatic cancer. Insulin resistance emerges as a possible risk factor for both primary liver cancer and the mortality associated with chronic liver disease [8]. Moreover, it is significantly correlated with the development of HCC [9], particularly in individuals with chronic hepatitis C virus infection [10]. Furthermore, a higher prevalence of both NAFLD and insulin resistance was found in bladder cancer [11], implying that NAFLD and insulin resistance could serve as potential evidence impacting not only hepatic cancer but also other cancerous diseases.

Lifestyle modifications, such as reducing calorie intake, exercise, and weight loss are recommended for the treatment of NAFLD/NASH [12,13,14,15]. However, there are few effective therapies to address the increasing global prevalence of NAFLD/NASH [16]. Time-restricted feeding (TRF) is a dietary intervention that involves consuming food within a consistent daily time window, typically spanning 8 to 10 h, without the need to reduce overall calorie intake [17]. Several studies have reported favorable outcomes associated with TRF, including reductions in body weight and adiposity, improvements in blood pressure, and enhanced insulin sensitivity in both human and rodent studies [18,19,20]. While the precise mechanisms remain unexplored, it has been shown that TRF prevents the development of liver steatosis and NASH when fed a high-fat diet [18] or a high-fat high-sucrose diet [21].

The methionine- and choline-deficient diet (MCDD), commonly used as a dietary model for studying non-alcoholic steatohepatitis (NASH) in mice, contains a substantial amount of sucrose and fat (45% sucrose and 10% fat) while lacking methionine and choline (Research diet, A02082002BR). Methionine, an essential amino acid, plays a crucial role as an intermediate in synthesizing vital antioxidants, namely, S-adenosylmethionine (SAM) and glutathione (GSH) [22,23]. Choline is essential for de novo synthesis of phosphatidylcholine, necessary for exporting triglycerides (TG) through very low-density lipoprotein (VLDL) packaging [24]. Additionally, choline significantly contributes to maintaining the integrity of mitochondrial membranes; its deficiency leads to alterations in mitochondrial membrane composition, resulting in disruptions to mitochondrial bioenergetics and the process of fatty acid β-oxidation [25]. Consequently, deficiencies in methionine and choline contribute to hepatic steatosis, oxidative stress, inflammation, and fibrosis [26,27]. MCDD-fed rodents exhibit excessive steatosis and inflammation as early as 2 weeks after the diet onset, with a substantial elevation of serum alanine aminotransferase [28,29]. However, the mechanisms underlying MCDD-induced NASH mostly differ from those in human NASH pathogenesis, and this model exhibits reduced body weight and an absence of insulin resistance [30,31]. Nevertheless, the MCDD model has the advantage of effectively inducing severe hepatic steatosis and inflammation in animals, closely mirroring human NASH histology within a relatively shorter time than other dietary models of NASH [32].

Inositol polyphosphate multikinase (IPMK) is responsible for producing inositol tetrakisphosphate (IP4) and inositol pentakisphosphate (IP5) [33]. Additionally, IPMK exhibits PI3K activity [34] and plays a pivotal role in various cellular signaling pathways. IPMK functions as an adaptor protein, interacting with key proteins, including a mechanistic target of rapamycin (mTOR), AMP-activated protein kinase (AMPK), serum response factor (SRF), and Unc-51-like autophagy-activating kinase (ULK) [35,36,37,38]. Recent research has uncovered IPMK‘s significant role as a key regulator of Toll-like receptor (TLR) signaling, particularly in myeloid cells, especially macrophages [39,40]. In this context, IPMK modulates the production of inflammatory cytokines by regulating TRAF6 protein stability through direct interaction with TRAF6 and preventing K48-linked ubiquitination. IPMK has been shown to play a critical role in liver inflammation [37]. We have reported that IPMK enhances hepatic insulin sensitivity and suppresses gluconeogenesis [41]. In preliminary studies, we found that TRF increased IPMK expression in liver. 

In this study, we show that time-restricted feeding (TRF) improves steatosis and inflammation in a mouse model of NASH, i.e., mice fed a methionine–choline deficient diet (MCDD). We found that IPMK protein was decreased in the liver of MCDD mice. To determine whether the loss of IPMK exacerbates the NASH phenotype, we fed wild-type (WT) and liver-specific IPMK knockout (LKO) mice an MCDD, and assessed liver histology, hepatocellular injury, and inflammation. Hepatic IPMK deficiency worsened hepatic steatosis, elevated serum ALT and AST levels, and expression of inflammatory genes in mice fed MCDD, suggesting a role as a molecular mediator in the effect of TRF in NASH.

## 2. Results

### 2.1. IPMK Is Decreased in the Liver of NASH Mice

We previously reported that IPMK protein level was decreased in mice fed a high fat diet (HFD) and the absence of hepatic IPMK led to an increase in body weight compared to WT mice under HFD [41]. To determine whether hepatic IPMK expression is affected during NASH progression, we measured IPMK protein levels in the liver of C57BL/6J mice fed MCDD. Similar to findings from other studies [42,43], the MCDD significantly reduced body weight (Figure 1A) and increased hepatic TG and serum ALT and AST (Figure 1B–D). These changes were consistent with the observed increase in hepatic steatosis and inflammation as shown in the histological analysis (Figure 1E). H&E staining revealed an increase in lipid droplets (black arrows) and inflammatory cell infiltration (red arrows) in the liver of MCDD mice compared to ND mice. Additionally, there was an increase in Masson’s trichrome and Sirius red staining (blue arrows) in MCDD mice, suggesting an elevated presence of fibrosis (Figure 1E). 

The expression levels of pro-inflammatory cytokines, such as tumor necrosis factor-α (TNFα), showed a significant increase. Additionally, there was a significant upregulation of pro-inflammatory chemokine genes such as monocyte chemoattractant protein-1 (MCP1), C–C motif chemokine ligand CCL3, CCL4, and CCL5. Moreover, genes associated with fibrosis, including collagen type 1 α1 (COL1α1), tissue inhibitor matrix metalloproteinase 1 (TIMP1), and transforming growth factor β (TGFβ), were also upregulated in the livers of MCDD mice (Figure 2A–L). Notably, both IPMK protein levels (Figure 1F) and mRNA expression (Figure 2A) were significantly lower in the liver of MCDD mice compared to mice on the control diet (ND-fed mice).

### 2.2. Loss of Hepatic IPMK Exacerbates NASH Progression

Since IPMK expression was significantly reduced in the liver of MCDD WT mice, we investigated whether IPMK is involved in NASH progression in a liver-specific IPMK knockout (LKO) mouse model. WT and LKO mice were fed MCDD for a short duration, 2 weeks, to induce steatohepatitis. Both the WT and LKO mice lost similar amounts of body weight (Figure 3A); however, hepatic TG and serum ALT and AST levels were significantly increased in the LKO mice compared to the WT mice (Figure 3B–D). Histological analysis revealed enlarged lipid droplet areas (black arrows), enhanced inflammatory cell infiltration (red arrows), and an increased presence of collagen fiber (blue arrows) in the livers of LKO mice on MCDD. This suggests heightened hepatic steatosis, inflammation, and fibrosis in LKO mice on MCDD compared to WT (Figure 3E).

The expression levels of pro-inflammatory cytokines were not significantly different between the WT and LKO mice (Figure 4B–D). However, pro-inflammatory chemokines (MCP1, CCL3, CCL4, and CCL5) showed a significant increase in the LKO liver of MCDD mice (Figure 4E–H). Although there was no significant difference, fibrosis-related genes such as COL1α1, TIMP-1, and TGFβ were also increased in the LKO liver of MCDD mice (Figure 4I–L). These results suggest that hepatic IPMK deficiency promotes NASH progression.

### 2.3. TRF Prevents NASH Progression and Restores IPMK Expression Level

Many studies have shown that TRF has beneficial metabolic effects such as improved insulin sensitivity, reduced hepatic steatosis and hyperlipidemia, and amelioration of oxidative stress and inflammation [44,45,46]. We determined whether TRF could prevent the NASH phenotype induced by MCDD. To investigate the effects of TRF on NASH, 10-week-old mice were fed MCDD ad libitum (ALF) or were restricted to feeding from 7 p.m. to 8 a.m. (TRF) for 2 weeks. Body weight was reduced in the TRF group compared to the ALF group (Figure 5A), despite no significant change in food intake (Figure 5F), and hepatic TG and serum ALT and AST levels were reduced compared to the ALF mice (Figure 5B–D). Histological analysis showed decreased hepatic steatosis, inflammation, and fibrosis in the liver of the TRF mice (Figure 5E).

The hepatic expression levels of pro-inflammatory cytokines, such as TNFα, interleukin (IL)-1β, and IL-6 (Figure 6B–D), along with chemokines including MCP1, CCL3, CCL4, and CCL5 (Figure 6E–H), exhibited a significant decrease in mice subjected to time-restricted feeding (TRF). Furthermore, a marked reduction was observed in the expression of fibrosis-related genes, specifically COL1α1 and TIMP1 (Figure 6I,K), in the liver of TRF mice. While the mRNA level of IPMK did not differ, there was a tendency for an increase in IPMK protein levels in the liver of the TRF mice compared to ALF mice (Figure 5G). These results suggest that TRF prevents NASH progression by maintaining hepatic IPMK levels.

## 3. Discussion

Non-alcoholic fatty liver disease (NAFLD) is a major public health challenge, affecting 20–30% of adults in the general population, and more than 70% of patients with obesity and diabetes [47]. NAFLD may progress from simple steatosis to NASH, cirrhosis, liver failure, and hepatocellular carcinoma [48]. Dietary calorie restriction reduces weight and hepatic steatosis in people with NAFLD; however, the long-term adherence to calorie restriction is challenging. Time-restricted eating (TRE) is a popular intermittent fasting regimen involving a specific eating period within a 24-hour cycle. TRE has gained attention because it is easier to adhere to and has metabolic benefits [49]. Studies in rodents suggest that food timing rather than total calorie intake per se underlies the beneficial effects of TRE [18]. Although TRF reduces hepatic lipids and inflammation in mice, the underlying mechanisms are unknown [18,46]. 

In this study, we demonstrated a mechanistic link between TRF, IPMK, and NASH in a mouse model. Hepatic-specific deletion of IPMK exacerbates MCDD-induced steatohepatitis and fibrosis in mice. TRF prevents a decrease in IPMK expression by MCDD and attenuates MCDD-induced steatohepatitis. Many groups have examined the effects of TRF on metabolic parameters such as glucose regulation, insulin sensitivity, and body composition, in animals and humans [18,19,20]. The initial evidence supporting the metabolic benefits of TRF was gathered through experiments conducted on mice with diet-induced obesity, utilizing a high-fat diet [18]. Subsequent research has consistently used diets designed to induce obesity, including high-fat diets, high-fat plus high-sucrose diets, and high-fructose diets, in order to investigate the effects of TRF. However, no study to date has investigated the effects of TRF in the context of NASH induced by MCDD. Although MCDD does have its limitations and impacts other tissues, including weight loss and the absence of insulin resistance, increased lipolysis in adipose tissue, intestinal inflammation, and impaired hippocampal neurogenesis, it has been a commonly used dietary model for NASH in mice, which results in liver injury characterized by excessive fat accumulation, inflammation, and fibrosis [50,51,52,53,54,55]. In our study, we observed that TRF significantly reduced hepatic lipid accumulation and liver damage caused by the MCDD. Interestingly, despite the MCDD being a diet that promotes rapid weight loss, the TRF group actually showed greater weight reduction compared to the ad libitum-fed (ALF) group (Figure 5A). This result is consistent with the weight reduction typically observed in TRF studies in mice [18,56].

Chemokines play an important role in maintaining tissue homeostasis and promoting inflammation in various organs, including the liver. Chemokines guide immune cells to the liver through specific chemokine receptors. Both liver-resident and infiltrating cells can release chemokines and cytokines in response to injury. Several chemokines are upregulated in liver and adipose tissues of patients with NAFLD [57]. MCP1, produced by hepatocytes, stellate cells, and Kupffer cells during liver pathology, is a crucial chemokine that stimulates the activation of monocytes and macrophages [58]. Elevated levels of MCP1 have been linked to the progression of hepatic inflammation and fibrosis [59]. CCL5 is another important chemokine in liver fibrogenesis [60]. Patients with liver cirrhosis often exhibit increased levels of CCL5, along with related chemokines CCL3 and CCL4 [60,61,62]. In our MCDD NASH model, we showed that TRF significantly reduced inflammatory chemokines (Figure 6E–H) as well as inflammatory cytokine gene expression (Figure 6B–D) compared to an ad libitum feeding group. Further investigation is necessary to evaluate how TRF specifically reduces chemokine gene expression induced by MCDD. It is possible that TRF modifies leukocyte responsiveness and improves inflammation [63,64].

Despite numerous studies on the beneficial effects of TRF, the molecular mechanisms have not been clearly elucidated. Many studies have focused on the role of circadian clock genes in metabolic architecture since feeding and fasting responses interact with the molecular circadian clock to affect diurnal rhythms [65,66,67,68]. However, Chaix et al. reported that TRF still prevents obesity and metabolic syndrome in mice lacking a circadian clock, such as Cry1:Cry2, Bmal1, and Rev-erva/b [69]. IPMK plays a crucial role in the metabolism of inositol polyphosphates (IPs) and functions as a versatile signaling factor intricately intertwined with essential biological pathways, including those related to AMPK, mTOR, and Akt [34,35,38]. Additionally, IPMK has been associated with liver inflammation [37] and oxidative stress [70]. Given that the development of NAFLD/NASH is linked to factors such as insulin resistance, lipid metabolism, oxidative stress, and inflammation, it is conceivable that IPMK may play a role in the development of NAFLD/NASH. Indeed, we found that IPMK expression was decreased by MCDD, and the loss of hepatic IPMK worsened steatohepatitis induced by MCDD. Interestingly, we found that TRF partially restored the MCDD-induced reduction in IPMK protein and significantly improved pro-inflammatory chemokine gene expression, suggesting that IPMK may play a causal role in the progression of NAFLD/NASH.

Our study has several limitations. We used the MCDD-induced steatohepatitis model to investigate the role of IPMK in NASH progression and the beneficial effects of TRF on NASH. However, many studies have reported severe weight loss in mice induced by MCDD. Similarly, in our study, we observed a substantial weight loss due to MCDD exposure (Figure 1A, Figure 3A and Figure 5A). Additionally, rodents did not develop insulin resistance under this dietary model. It is noteworthy that most humans with NASH are obese and insulin resistant, highlighting significant differences between MCDD-induced NASH models and human NASH. Despite IPMK being a key protein in our study, the precise mechanisms through which it regulates hepatic lipid metabolism and inflammation in MCDD-induced steatohepatitis remain unclear. Nevertheless, our study contributes valuable insights into the role of IPMK in NASH progression and the positive impact of TRF using the MCDD-induced steatohepatitis model.

Further studies are needed to understand how IPMK mediates changes in hepatic lipid metabolism, inflammation, and fibrosis in NASH, and how TRF modulates the levels of IPMK in the liver to prevent the development of NASH. Our study in MCDD mice presents a model for elucidating the pathophysiology and potential treatment of NAFLD/NASH.

## 4. Materials and Methods

### 4.1. Animals

Experiments were performed in accordance with the Institutional Animal Care and Use Committee guidelines with its approvals. The LKO mice were generated by crossing Albumin-CreTg/+ mice (Jackson Laboratories) with mice homozygous for a “floxed” exon 6 of IPMK (IPMK fl/fl). The control mice for this study were IPMK fl/fl (WT) mice. Mice were housed under standard conditions in a temperature- and humidity-controlled facility with a light–dark cycle of 12 h (lights on at 07:00) and fed ad libitum at the Johns Hopkins University (Baltimore, MD, USA). 

Experiment 1: Eight-week-old WT mice were fed an ND (A02082003BY, Research Diets, New Brunswick, NJ, USA) or MCDD (A02082002BR, Research Diets, New Brunswick, NJ, USA) for 2 weeks.

Experiment 2: Eight-week-old WT and LKO mice were fed MCDD for 2 weeks.

Experiment 3: Ten-week-old WT mice were divided into two groups. One group (ALF) had ad libitum access to the MCDD diet, while the other group underwent time-restricted feeding (TRF), with access to the MCDD diet from 7 p.m. to 8 a.m., for two weeks.

All mice were sacrificed after a 5 h fast (from 08:00 a.m. to 1:00 p.m.), and blood was collected from the heart before harvesting liver tissues.

### 4.2. Liver Triglyceride Measurement

Liver triglyceride was extracted using the Folch extraction method [71], and the triglyceride levels were measured using the Stanbio Triglyceride LiquiColor kit (Stanbio Laboratory, Boerne, TX, USA) according to the manufacturer’s protocol.

### 4.3. ALT and AST Measurement

Mice were euthanized and blood was obtained from cardiac puncture and centrifuged (2000× *g*, 10 min) at 4 °C. The serum layer was collected and stored at −80 °C. Serum ALT and AST were measured using the ALT/SGPT Liqui-UV^®^ test and AST/SGOT Liqui-UV^®^ test (Stanbio Laboratory, Boerne, TX, USA) according to the manufacturer’s protocol.

### 4.4. Liver Histology

Liver tissues fixed with PBS containing 4% paraformaldehyde were embedded in paraffin, and 5 μm sections were stained with hematoxylin and eosin solution, Masson’s trichrome solution or Sirius red solution, according to standard procedures.

### 4.5. Immunoblotting

Liver tissues were lysed in RIPA buffer (50 mM Tris (pH 7.4), 150 mM NaCl, 1% Triton X-100, 15% glycerol) containing inhibitors (phosphatase inhibitor mixture 2 and 3 (MilliporeSigma, Burlington, MA, USA), and protease inhibitor mixture 1 and 2 (Roche Applied Science, Mannheim, Germany) and heated at 95 °C for 5 min prior to electrophoresis. Equivalent amounts of protein (20 µg) in sodium dodecyl sulfate (SDS) sample buffer (50 mM Tris–HCl at pH 6.8, 2% SDS, 100 mM DL-dithiothreitol, 10% glycerol) were separated by 4–12% SDS–polyacrylamide gel electrophoresis and then transferred to a 0.2 mm nitrocellulose membrane, blocked with 5% nonfat dry milk, and incubated with primary antibodies at 4 °C overnight. Immunoblotting was conducted with the following antibodies: IPMK from Novus Biologicals (NBP1-32250, Centennial, CO, USA), GAPDH from Cell Signaling Technology (2118S, Danvers, MA, USA). Blots were imaged and quantitated using an Odyssey Near-Infrared Scanner (Li-Cor Biosciences, Lincoln, NE, USA). 

### 4.6. Quantitative Real-Time PCR

Total RNA was isolated from WT or LKO mouse liver tissue or primary hepatocytes using TRIZol reagent (Thermo Fisher Scientific, Waltham, MA, USA), and chloroform (MilliporeSigma, Burlington, MA, USA), precipitated in 2-propanol (MilliporeSigma, Burlington, MA, USA), washed in ethanol (MilliporeSigma, Burlington, MA, USA), and quantified using Epoch Microplate Spectrophotometer (Agilent Technologies, Santa Clara, CA, USA). mRNA was reverse transcribed into cDNA using ProtoScript^®^ II First Strand cDNA Synthesis Kit (BioLabs, Ipswich, MA, USA). Ribosomal Protein L32 (RPL32) mRNA was used as the invariant endogenous control and melting curve analysis was run to verify specificity of each amplicon. The relative amounts of the RNAs were calculated using the comparative threshold cycle method. Primers were as follows: IPMK forward 5′-TGC CCA ACT GGA AAA GGT GT-3′, reverse 5′-CCC TCA TCG ACT GTG TTG CT-3′; TNFα forward 5′-TCG TAG CAA ACC ACC AAG TG-3′, reverse 5′-AGA TAG CAA ATC GGC TGA CG-3′; IL-1β forward 5′-TCT CGC AGC AGC ACA TCA ACA-3′, reverse 5′-CCT GGA AGG TCC ACG GGA AA-3′; IL-6 forward 5′-GAC CTG TCT ATA CCA CTT CAC-3′, reverse 5′-GTG CAT CAT CGT TGT TCA TAC-3′; MCP-1 forward 5′-AGG TCC CTG TCA TGC TTC TG-3′, reverse 5′-TCT GGA CCC ATT CCT TCT TG-3′; CCL3 forward 5′-ACT GCC CTT GCT GTT CTT CT-3′, reverse 5′-GTC TCT TTG GAG TCA GCG CA-3′; CCL4 forward 5′-CTC TCT CCT CTT GCT CGT GG-3′, reverse 5′-CTC ACT GGG GTT AGC ACA GA-3′; CCL5 forward 5′-CCA TCA TCC TCA CTG CAG CC-3′, reverse 5′-CTC TGG GTT GGC ACA CAC TT-3′; COL1α1 forward 5′-GAG CGG AGA GTA CTG GAT CG-3′, reverse 5′-GCT TCT TTT CCT TGG GGT TC-3′; ACTA2 forward 5′-CTG ACA GAG GCA CCA CTG AA-3′, reverse 5′-CAT CTC CAG AGT CCA GCA CA-3′; TIMP-1 forward 5′-TCC CTT GCA AAC TGG AGA GT-3′, reverse 5′-GAC GGC TCT GGT AGT CCT CA-3′; TGFβ forward 5′-TTG CTT CAG CTC CAC AGA GA-3′, reverse 5′-TGG TTG TAG AGG GCA AGG AC-3′; Rpl32 forward 5′-AAG CGA AAC TGG CGG AAA CC-3′, reverse 5′-CCC ATA ACC GAT GTT GGG CA-3′.

### 4.7. Statistical Analysis

Statistical analyses were performed using GraphPad Prism 10 (GraphPad, San Diego, CA, USA). Data are presented as mean ± SEM. Comparisons between two groups were carried out using Student’s *t*-test. The threshold for statistical significance was set at *p* < 0.05. 

## Figures and Tables

**Figure 1 ijms-25-01390-f001:**
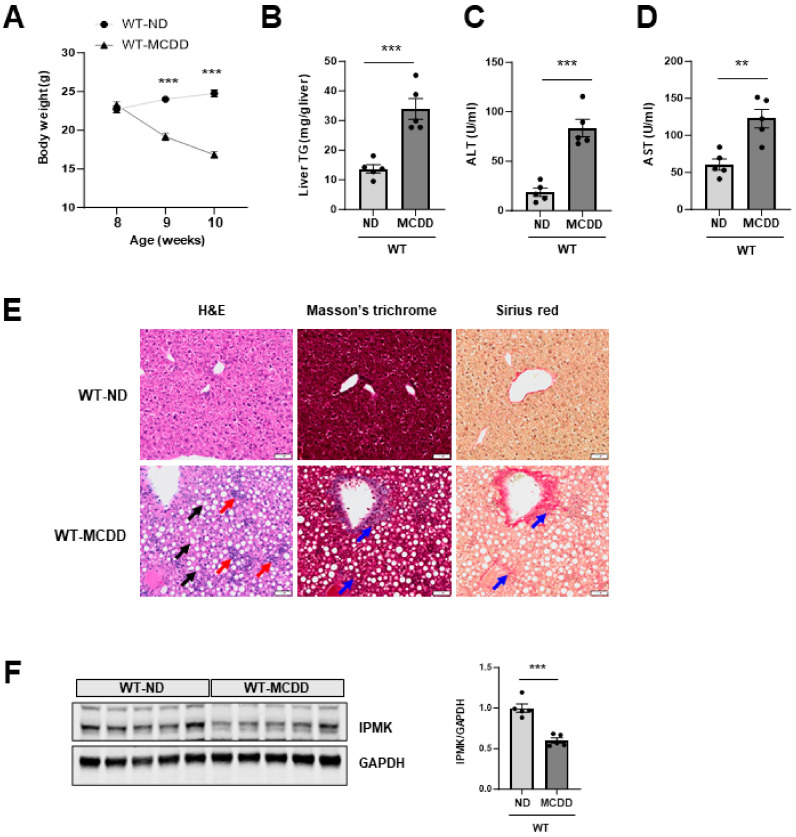
IPMK protein is decreased in the liver of MCDD mice. (**A**) Body weight of ND- and MCDD- fed mice (*n* = 5 for both) was measured weekly from age 8 to 10 weeks. (**B**) Liver TG, serum (**C**) ALT and (**D**) AST were measured to evaluate the levels of liver injury. (**E**) Liver paraffin sections were stained with H&E, Masson’s trichrome, and Sirius red to determine the levels of steatosis, inflammation, and fibrosis. The black arrow indicates the region of a lipid droplet. The red arrow indicates the area with inflammatory cell infiltration. The blue arrow indicates the region characterized by fibrosis (scale bar, 50 µm). (**F**) Protein levels of IPMK in the liver from ND- or MCDD-fed mice were analyzed by immunoblotting (*n* = 5 per group). Levels of IPMK protein were quantified. Data are mean ± SEM; ** *p* < 0.01, *** *p* < 0.001.

**Figure 2 ijms-25-01390-f002:**
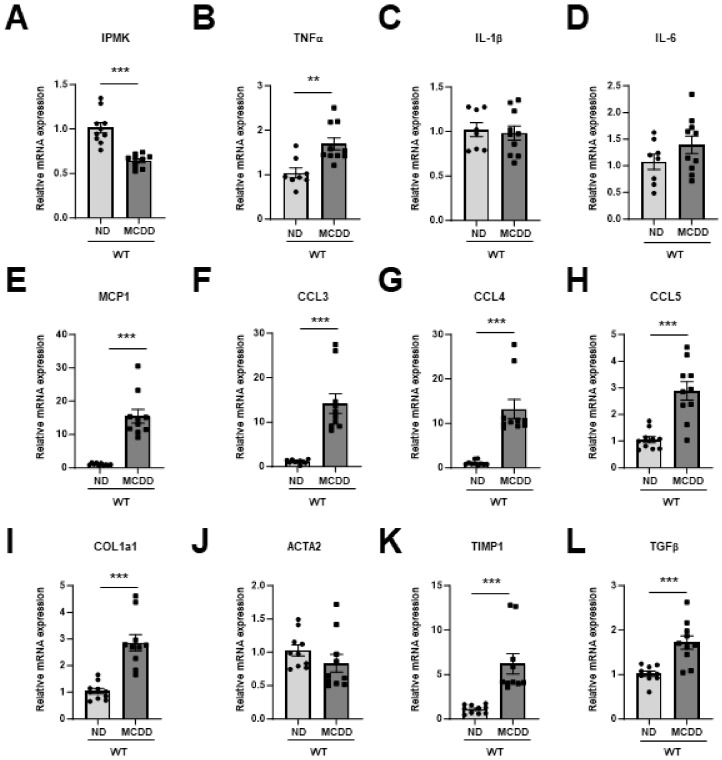
IPMK mRNA is decreased in the liver of MCDD mice. mRNA levels of (**A**) IPMK, (**B**–**D**) inflammatory cytokines (TNFα, IL-1β, and IL-6), (**E**–**H**) chemokines (MCP1, CCL3, CCL4, and CCL5), and (**I**–**L**) fibrosis-related genes (COL1α1, smooth muscle and aorta actin α-2 (ACTA2), TIMP1, and TGFβ) in the liver from ND and MCDD-fed mice (*n* = 4–5 per group). Data are mean ± SEM; ** *p* < 0.01, *** *p* < 0.001.

**Figure 3 ijms-25-01390-f003:**
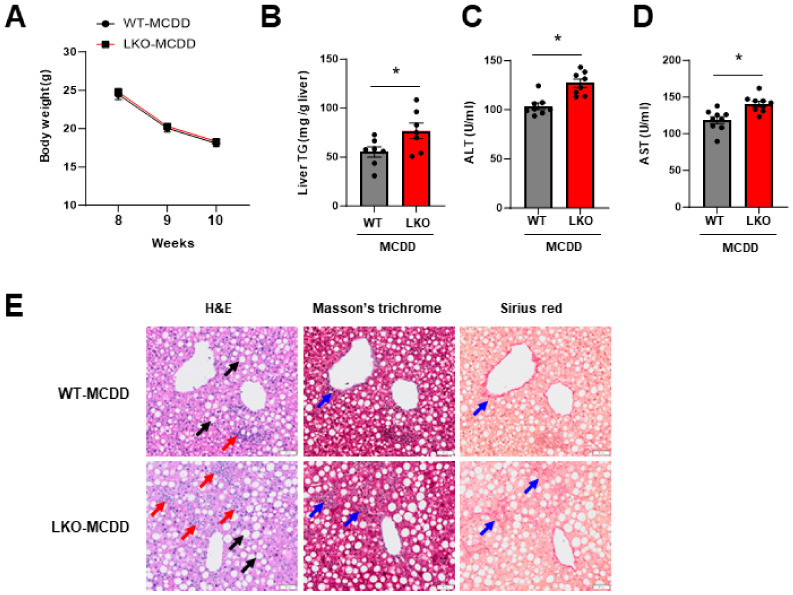
Loss of hepatic IPMK exacerbates NASH progression. (**A**) Body weight of WT (*n* = 5–8) and LKO (*n* = 5–9) mice fed MCDD was measured weekly from age 8 to 10 weeks. (**B**) Liver TG, serum (**C**) ALT and (**D**) AST were measured to evaluate the levels of liver injury. (**E**) Liver paraffin sections were stained with H&E, Masson’s trichrome, and Sirius red to determine the levels of steatosis, inflammation, and fibrosis. The black arrow indicates the region of a lipid droplet. The red arrow indicates the area with inflammatory cell infiltration. The blue arrow indicates the region characterized by fibrosis (scale bar, 50 µm). Data are mean ± SEM; * *p* < 0.05.

**Figure 4 ijms-25-01390-f004:**
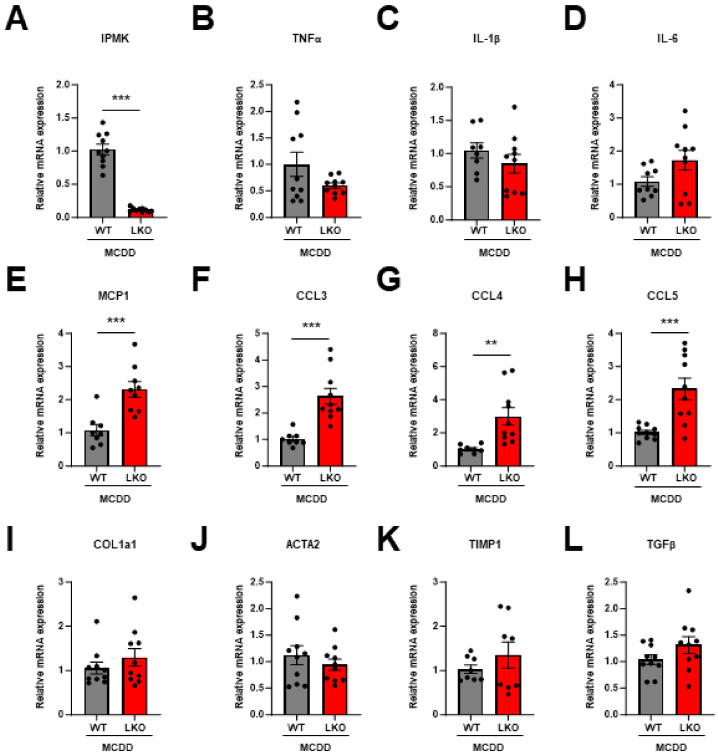
Loss of hepatic IPMK upregulates pro-inflammatory chemokine gene expression. mRNA levels of (**A**) IPMK, (**B**–**D**) inflammatory cytokines (TNFα, IL-1β, and IL-6), (**E**–**H**) chemokines (MCP1, CCL3, CCL4, and CCL5), and (**I**–**L**) fibrosis-related genes (COL1α1, ACTA2, TIMP1, and TGFβ) in the liver from MCDD WT and LKO mice (*n* = 4–5 per group). Data are mean ± SEM; ** *p* < 0.01, *** *p* < 0.001.

**Figure 5 ijms-25-01390-f005:**
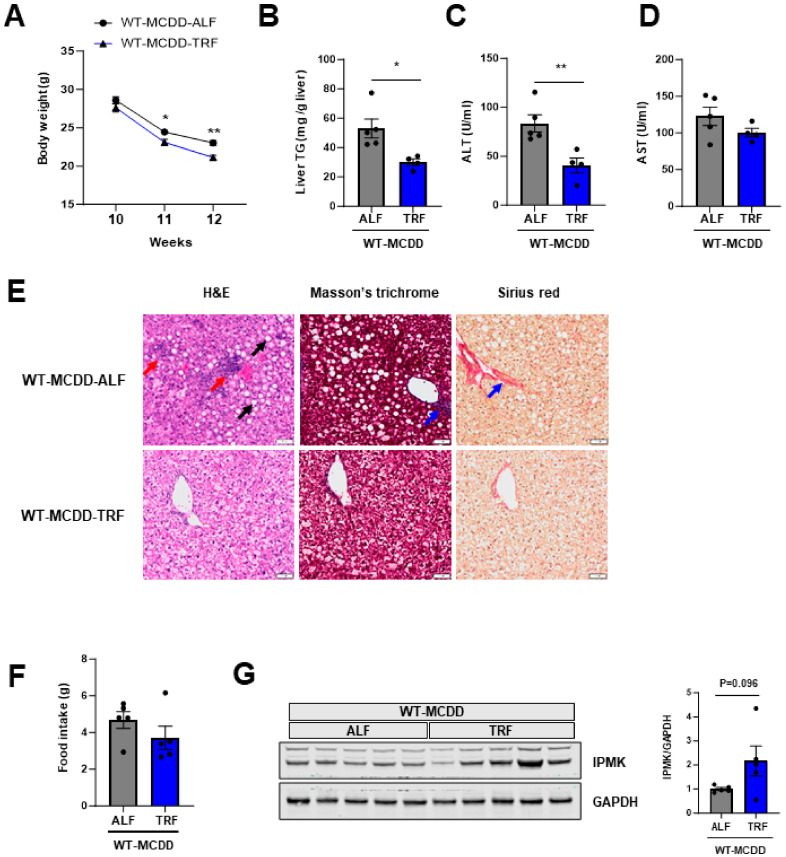
TRF prevents NASH progression. (**A**) Body weights of ad libitum-fed (ALF) (*n* = 5) and time-restricted-fed (TRF) (*n* = 4–5) mice on MCDD were measured weekly from age 10 to 12 weeks. (**B**) Liver TG, serum (**C**) ALT and (**D**) AST were measured to evaluate the levels of liver injury. (**E**) Liver paraffin sections were stained with H&E, Masson’s trichrome, and Sirius red to determine the levels of steatosis, inflammation, and fibrosis. The black arrow indicates the region of a lipid droplet. The red arrow indicates the area with inflammatory cell infiltration. The blue arrow indicates the region characterized by fibrosis (scale bar, 50 µm). (**F**) Food intake (g/day) of ALF and TRF mice fed MCDD was measured. (**G**) Protein levels of IPMK in the liver from ALF and TRF mice were analyzed by immunoblotting (*n* = 5 per group). Levels of IPMK protein were quantified. Data are mean ± SEM; * *p* < 0.05, ** *p* < 0.01.

**Figure 6 ijms-25-01390-f006:**
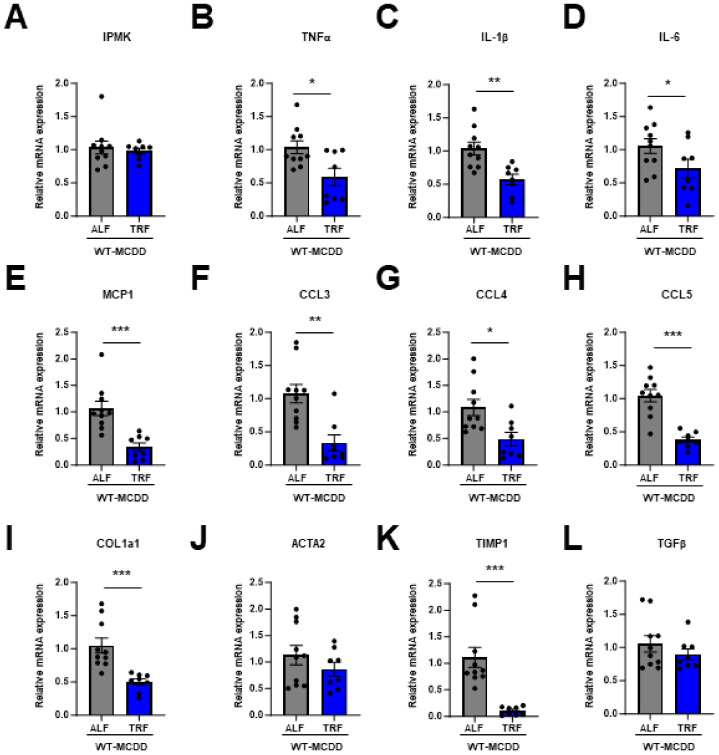
TRF prevents MCDD-induced pro-inflammatory gene expression. mRNA levels of (**A**) IPMK, (**B**–**D**) inflammatory cytokines (TNFα, IL-1β, and IL-6), (**E**–**H**) chemokines (MCP1, CCL3, CCL4, and CCL5), and (**I**–**L**) fibrosis-related genes (COL1α1, ACTA2, TIMP1, and TGFβ) were measured in the livers of both the ALF and TRF groups (*n* = 4–5 per group). Data are mean ± SEM; * *p* < 0.05, ** *p* < 0.01, *** *p* < 0.001.

## Data Availability

The data presented in this study are openly available in bioRxiv at https://doi.org/10.1101/2023.11.15.567214 (accessed on 17 November 2023).

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
