# Peer review of "Time-Restricted Feeding Ameliorates Methionine–Choline Deficient Diet-Induced Steatohepatitis in Mice"

_ijms, 2024, doi:10.3390/ijms25031390_

Round 1

Reviewer 1 Report

Comments and Suggestions for Authors

An interesting paper, the experimental work behind these results is notable; however, there remain many doubts that I would like to ask:

What do the authors know about the current name, Metabolic Dysfunction-Associated Steatotic Liver Disease (MASLD), from the formerly called non-alcoholic steatohepatitis (NASH)?

What do the authors think of using an MCDD model of hepatic steatosis without metabolic dysfunction and body weight loss? Please discuss possible toxicity to other organs or systems.

 The figures corresponding to histology are unclear; larger and better-resolution images would help to see the differences. It is not possible to judge them that way.

Figure 1F they have multiple bands, and the one they consider to correspond to IPMK has a double band, how do they identify the correct band? As an essential element in evaluating this work, using a more specific antibody would be convenient. What antibody was used? Is it monoclonal or polyclonal? Could you please share the complete images of the membrane?

Why do you use 8-10 weeks to cause hepatic steatosis when characterizing the MCDD model and subsequently only use two weeks?

Please explain ¿why in Figure 4B, the TNFa WT group is not normalized to 1? And why doesn't LKO show an increase? On the contrary, it decreases (although not significantly)

It needs to be clarified because the MCDD group (8-10 weeks) has average TG concentrations of 35mg/gliver while WT (2 weeks) has average concentrations of 50mg/gliver; there is also a discrepancy in ALT and AST. It is considering that they have less exposure time to the diet.

Could you explain why there is no greater expression of fibrosis markers with LKO? Does it prevent fibrosis?

How do you consider TRF intervention when the same MCDD diet causes body weight loss? There is no justification to say that losing more weight is beneficial for the body. I insist that there are more appropriate diets for studying hepatic steatosis with metabolic dysfunction. The authors explain that TRF has metabolic benefits and improves insulin sensitivity, but they do not explore any of this.

Could they be more precise in the TRF group? ¿Are these animals with MCDD-induced steatosis separated into two groups, one with free access to the diet and the other with restriction? Or are they healthy animals that start with a free and restricted diet? If so, is the ALF group the equivalent of WT? Review ALT and AST concentrations among these groups.

Why are there no significant differences in intake if there was restriction? Why, if there is no decrease in intake, are there changes in TG, AST, and ALT?

The 5G graph has a value of p=0.096, but in no other condition does it have a value of P, especially if it is not significant.

In discussion:

They mention: Studies in rodents suggest 158 ​​that food timing rather than total calorie intake per se underlies the beneficial effects of 159 TRE [14], but it was not demonstrated that there was a decrease in total intake.

They also justify the study with obesity and metabolic dysfunction, but their model does not have these characteristics.

I suggest that the groups be identified with equivalent nomenclature and colors since they generate confusion and appear to be different groups when they are not.

Note: I understand there is information in the literature on using the MCDD model. Still, it is crucial to be critical and know that if we intend to generate models that help understand human mechanisms, we must try to use models closer to reality, such as other diets well-documented high fat. From my point of view, this model is a way to generate steatosis, but it does not emulate reality in terms of metabolism. In my opinion, it is not the best model to study MASLD.

Thank you for considering my observations.

Reviewer 2 Report

Comments and Suggestions for Authors

The authors analyze the influence of time-restricted feeding (TRF) on the clinical picture of non-alcoholic steatohepatitis (NASH), with a focus on inositol polyphosphate multikinase (IPMK). The study shows that the deletion of hepatic IPMK leads to the aggravation of methionine-choline-deficient diet-induced NASH. Additionally, TRF improves the biomarkers of NASH and improves the clinical picture.

Overall impression of the work: favorable.

Introduction: The authors have provided all necessary references and the writing is clear and concise. However, they could provide more detail on the MCDD diet. 

Important general remarks: improving the quality of the provided pictures is necessary. The figures appear out of focus, with certain difficulties to read the axis title, for example, which decreases the overall appeal of this interesting work. This especially applies to the liver histology pictures and blots. A discussion (or even a mention) of effect sizes (beyond statistics) would also be very helpful (especially as regards Fig. 3B-D). The statement in line 211, "... we found that TRF restored the MCDD-induced reduction in IMPK protein..." is not supported by Figures 5G and 6A, which show that TRF does not affect the expression or transcription of the protein. Although the increase in effect size can be recognized in the protein expression analysis (Fig. 5G), it is far from being significant and therefore the statement in lines 211 -212 should be toned down. In a few other instances, the authors should also lower the strength of their statements, which appear too strong when compared to the only modest biological effects. Please check your manuscript accordingly.

In section 4.1, please specify the exact time of animal sacrifice, as it is relevant for various dietary interventions and may affect biomarkers. Additionally, in section 4.5, please provide the lot/article numbers of the antibodies used.

Round 2

Reviewer 1 Report

Comments and Suggestions for Authors

I very much appreciate the answers to my questions.

I completely understand that you call your model NASH, due to all the deficiencies that the MCDD model has. However, it is necessary to consider that we seek to develop models that contribute to understanding human pathological processes, if not for what? The research must go in that direction, because otherwise, extrapolating these observations to a different dysmetabolic context (as happens in humans) may not be useful, since the mechanisms may be different.

How the results are presented, and the conclusions of the experiments seem to me to be reviewed in depth. The new version has repeated figures. The abbreviations change for the same experimental approach (MCDD=WT=ALF) and are confusing. Normalization is performed with ND and subsequently with WT or ALF, which does not allow us to see if the changes in TRF lead to normality. It seems to me that integrating the data would help understand the differences between the groups.

Thank you for the question. In the study by Rinella et al., the term metabolic dysfunction- associated steatotic liver disease (MASLD) was chosen to replace non-alcoholic fatty liver disease (NAFLD) after four rounds of Delphi survey. A consensus was reached to alter the definition, now requiring the presence of at least one of five cardiometabolic risk factors. Conceptually, individuals previously categorized under NAFLD can now be considered fully covered by the categories of MASLD and potential MASLD. Additionally, by maintaining the term and clinical definition of steatohepatitis, data from past clinical trials and biomarker discovery studies involving NASH patients remain relevant and applicable to individuals classified as MASLD or MASH under the new nomenclature. This transition doesn't impede research efficiency (Rinella et al., J Hepatol, 2023). In this study, we were able to emphasize hepatic steatosis and steatohepatitis using MCDD. We believe that the MCDD model is not intended for a diet related to the new term Metabolic Dysfunction-Associated Steatotic Liver Disease (MASLD). Therefore, we chose to use the term NASH (Non-Alcoholic Steatohepatitis).

MCDD model not only has the absence of obesity but since the weight loss of the animals is substantial, taking a mouse from 25g to 16-17g is a substantial loss, just due to the intervention (at least 30% loss). The included text makes sense and notes that the important limitations of the study

·       Thank you for modifying the figures corresponding to histology, but they leave the interpretation up to the reader since they do not point out and describe them. They still seem small and low-resolution

·       According to the Novus Biologicals product datasheet (NBP1-32250) of the antibody for IPMK, a single band is obtained with cell lysate, with the understanding that these are not the same conditions with tissues, it seems to be due to technical problems that could be improved. The two gels that were sent to me do not show the same number of nonspecific bands and even the gel in Figure 1, has a band of approximately 55 KDa differentially expressed in the ND and MCDD groups. IPMK being the most important protein analyzed in the study, leaves too many doubts in the air, which could probably be reduced with experimental changes during the WB. Generally speaking, a WB with so many nonspecific bands is not acceptable. Question, was the optical density of the WB bands only carried out on the images that were sent to me? Was there any repetition?

·       Thanks for the explanation about the ages of the animals.

·       AST in ALF is approximately 70 U/ml lower than the range mentioned in your answer (110-120 U/ml)

·       Figure 4, says that loss of hepatic IPMK upregulates pro-inflammatory chemokine but TNFa, IL1b, and IL6 do not have statistically significant differences (WT vs LKO), so the statements are not supported by the results. In fact, IL1b nor IL6 seem to be modified in any of the experimental conditions.

·       Genes related to fibrosis (Col1a1, TIMP1, and TGFb) do not have significant differences between WT vs LKO, it is not correct to say, “there is a tendency”. Therefore, why conclude that IPMK deficiency promotes the progression of NASH if the evidence of fibrosis is not clear? There is also no description of the histology to confirm this.

·       You say “ Interestingly, we found that TRF partially restored the MCDD-induced reduction in IPMK protein and significantly improved pro-inflammatory chemokine gene expression, suggesting that IPMK may play a causal role in the progression of NAFLD/NASH

Partially? There are no significant differences in IPMK (ALF vs TRF). I think they do not have sufficient evidence that IPMK has a CAUSAL ROLE in the progression of NASH since in LKO there is no more fibrosis.

·       Thank you for your explanation of the model, although I differ in the statement, since this model does not increase weight (on the contrary substantially reduce it) we cannot speak of similarity with the human since the key elements are obesity and dysmetabolism. What I believe is that MCDD is a model only of liver damage, but it is NOT similar, nor should it be based on what we observe in humans.

·       We appreciate the insightful comments. We agree that MCDD diet does not replicate all the human pathology observed in NASH or MASLD. However, it does still capture certain features, such as increased inflammation and a certain degree of fibrosis. We are currently planning to explore more appropriate models to examine the effects of TRF on MASLD.

                     I absolutely agree

In conclusion, the writing requires a thorough review of the figures and being cautious with its statements.

Round 3

Reviewer 1 Report

Comments and Suggestions for Authors

I appreciate the modifications made to your writing. Thank you for your willingness to be clearer and more objective in highlighting the contributions of your work as well as its limitations. Greetings

Author Response

I appreciate the modifications made to your writing. Thank you for your willingness to be clearer and more objective in highlighting the contributions of your work as well as its limitations. Greetings

- We appreciate your thoughtful feedback. We are pleased that the modifications were well received. Your encouragement to be clearer and more objective is valued, and we are grateful for your recognition of the work's contributions and limitations.